# Basal Stimulation as Developmental Support in At-Risk Newborns: A Literature Review

**DOI:** 10.3390/children10020389

**Published:** 2023-02-16

**Authors:** Petra Potmesilova, Milon Potmesil, Jana Mareckova

**Affiliations:** 1Department of Christian Education, St. Cyril and Methodius Faculty of Theology, Palacký University Olomouc, 77900 Olomouc, Czech Republic; 2Center of Evidence-Based Education & Arts Therapies: A Joanna Briggs Institute Affiliated Group, Faculty of Education, Palacky University Olomouc, 77900 Olomouc, Czech Republic; 3Department of Anthropology and Health Education, Faculty of Education, Palacky University Olomouc, 77900 Olomouc, Czech Republic

**Keywords:** Basal Stimulation, cognitive–behavioral functions, temperament, preterm, disabled, infant

## Abstract

Background: The aim of this review of the literature was to find and summarize relevant research evidence available within the scientific sources and gray literature in accordance with the JBI recommendations. Search question: What effect does Basal Stimulation have on the cognitive–behavioral functions or temperament of a preterm or disabled infant? Methods: The following sources were searched: PSYCINFO, MEDLINE, PsycArticles, ERIC, Wiley Online Library, ProQuest Scopus, WOS, JSTOR, Google Scholar, and MedNar. The study contains an analysis of texts that have been published in the English, Czech, and German languages. The search time span was set at 15 years. Results: A total of 15 sources were found for the specified topic. Conclusions: In all cases, there was confirmation about the positive influence of the concept of “Basal Stimulation” on the cognitive–behavioral functions and temperament of premature and disabled children.

## 1. Introduction

A premature newborn is a baby born before the 38th week of gestation with a weight of less than 2500 g. The reasons for which preterm births occur are multifactorial and very often the cause is not even clarified. Preterm birth is defined as birth before the 37th week of gestation. Preterm birth is currently considered to be a syndrome where its etiology involves various factors leading to uterine activation and subsequent delivery. The incidence of preterm birth across countries according to the WHO [1] ranges from 5% to 18% of all babies born. Preterm birth complications are the leading cause of death among children under five years of age, which was responsible for approximately one million deaths in 2015 [2].

For this study, the synactive development theory [3,4] was chosen as the underlying theoretical approach. The author of the synactive theory of development is Heidelise Als. This theory expresses the continuous interaction and regulation of autonomic, motor, and behavioral subsystems in interaction with the environment [3]. The first natural environment for the fetus is the womb, where it is maintained in a comfortable environment that tightly confines the body (hence one of the parameters of Basal Stimulation—see later in the text). The fetus is exposed to tactile and motor stimuli from the amniotic fluid and amniotic envelope. The low level of gravity and movement in the liquid environment aids motor development and makes fetal movements gentler, calm, and subdued. Premature babies leave the quiet, dark, comfortable intrauterine environment too early and enter a world full of bright lights, noise, painful stimuli, and gravity and, above all, they are separated from the mother. In addition, various organs of the newborn may be exposed to the toxic effects of drugs, which can interfere with normal development [5]. In terms of the later development of the newborn child, the mother’s circadian rhythm is already important in pregnancy because it affects both the fetuses state of consciousness and its perception of sleep and wakefulness as it is stimulated by sensations and information from the mother. The synactive development theory [3] considers development as a multidimensional and multilayered process in which the two are intertwined and interact. Immature children, according to the synactive development theory, are viewed as fetuses developing in an evolutionarily unnatural extrauterine environment. The difference between the intrauterine environment, in which the fetal brain develops in the third trimester, and the environment outside the womb can significantly affect the child’s future development—neuro-physiologically, neuro-psychically, psycho-emotionally, and psycho-socially. Subsequently, the presence and exposure to other stimuli, such as pain and stress, which the immature organism cannot process, can alter the course of development with implications for lifelong health. Thus, the sensory stimulation of premature infants is one way to teach the child’s organism to respond to specific stimuli [6]. 

Overall, the premature separation of the fetus from the mother’s body is a burden that is difficult for the premature infant to cope with, and the stress caused affects the formation of cell membranes of neurons and other tissues and changes the architecture of brain structures. In the physiological newborn, birth stress activates protective systems, and the stress thus tolerated does not cause further consequences. In the immature newborn, exposure to external stressors leads to prolonged activation of the protective systems. This over-activated stress response can disrupt the architecture of the developing brain with potential long-term effects on cognitive functions, such as learning and behavior, as well as overall physical and mental health [7] p. 47. 

Developmental psychology in the twentieth century pointed to the gradual learning of coordination and integration of stimuli acquired by separate senses in infants. According to Piaget [8], it is not until the sixth month that the infant succeeds in integrating stimuli into more complex perceptions. The view of the development of the sensory organization is changing and more consideration is being given to a higher degree of plasticity linked to the acquired experiences of the physiological newborn. The situation is quite different for the preterm infant. It can be assumed that the normal stimulation of the senses, in this case, is beyond the acceptable limit, and overstimulation of hearing and vision is a source of strain and stress, and conversely for touch, there may be a reduction in perceptual sensitivity as a result of insufficient stimulus levels [9]. 

Already from an early age, the child’s development in the social–emotional sphere is manifested. Firstly, self-awareness is a manifestation of individual temperament and at the same time, the manifestation of temperament toward the social environment, which initially involves members of the immediate family. If the basis for development in the social–emotional domain is disturbed by a lack of stimuli and difficulties in sensory processing, which is the situation of the premature infant, significant developmental delays can be expected. Temperament as a unique innate characteristic of an individual is a regulator of activity, the intensity of emotions, mood alternation, and the ability to concentrate and flexibility is manifested from birth in physiological newborns [10]. In immature newborns, this manifestation of personality is suppressed by the negative influences mentioned above. Research on preterm infants who were assessed using the observational protocol of the Spanish version of the NIDCAP has shown that a reflection of stress levels and stability levels can be observed in variables focusing on free movements, body position, and an assessment of the newborn’s crying, type of sleep, wakefulness, facial expressions, etc. Premature babies can respond behaviorally to stress, and this response varies according to gestational age [11]. In response to stressful situations and stress in premature newborns, an individual rehabilitation approach can be programmed so that interaction with the environment—material and social—is set as early as possible and is used as a basis for the further development of the child. 

Basal Stimulation© (BS) is a rehabilitation concept that works on the pedagogical-treatment principle and allows support for the perception, communication, and physical abilities of a person with any disability, regardless of its type and severity. The BS concept is based on the assumption that the ability to receive stimuli as sensations is possible in any case [12,13]. The author of the BS concept is Andreas Fröhlich [14], who since 1970, worked as a special educator in the Westpfalz/Landstuhl rehabilitation center (Germany) with children who were diagnosed with severe physical and mental changes. On the basis of his practical experience, he developed the concept of BS, including its theoretical foundations, in collaboration with Christel Bienstein [15]. The concept entered rehabilitation practice as a rehabilitation method based on a comprehensive (holistic) approach to the patient. BS as a rehabilitation concept is aimed at patients regardless of their age. BS as a concept serves to initiate and develop physical, holistic learning. The application of BS, which is conceptualized as a holistic approach to stimulating development in the very early stages of life, is an effective tool to aid navigating new or unclear situations of one’s perception of the environment. Furthermore, it supports the development of communication and fine and gross motor skills. Another effect of BS is the reduction in stress in people in borderline stressful situations, e.g., severe medical conditions. This rehabilitation concept can also be characterized as psychotherapeutically oriented support in difficult phases of the perception and communication of clients. A comprehensive view of the client is the basis for therapeutic work in the BS concept, and a disturbed perception of the self and the environment is considered the main problem. Supporting the development of communication and the gradual development of social competencies are the most important target items. Since the process of the application of BS is about searching for and finding the client’s learning strategy, a positive influence on the development of the client’s cognitive functions and behavioral manifestations can be assumed. In situations where the client lacks sufficient exercise, has limited opportunities to communicate, and does not have enough environmental stimulation, there is a risk of demotivation and disorientation. The rehabilitation process is therefore, in its final form, a combination of physiotherapy and the consideration of psychological, educational, and special education therapy. BS can be characterized as a process of mutual learning between the client and therapist. The therapist seeks effective techniques that will bring about the first reactions—responses to stimulation—so that the spectrum of stimuli is both broadened and specified, which also brings about the development of client-specific responses. 

## 2. Materials and Methods

This literature review aimed to find and summarize the relevant research evidence available within scientific sources and gray literature. At first, a search question was formulated: 

What effect does Basal Stimulation (I) have on the cognitive and behavioral functions (O1) or temperament (O2) of a preterm or disabled infant (P)? 

This literature search was created because of the need to build a theoretical basis for a research project focused on the possibility of objective evaluation of the effectiveness of Basal Stimulation in children with disabilities under the age of three years by monitoring certain components of temperament. 

### Search Strategy

The following sources were searched: PsycINFO, MEDLINE, PsycArticles, ERIC, Wiley Online Library, ProQuest Scopus, WOS, JSTOR, Google Scholar, and MedNar. 

The study contains an analysis of texts published in the English, Czech, and German languages. A thesaurus was used to configure the keywords and phrases that were needed. 

The preliminary step was a search in the above-mentioned sources for the combination of keywords Basal Stimulation AND infant, as the goal was to find out whether relevant sources related to temperament or cognitive and behavioral functions affected by the therapeutic approach of Basal Stimulation focused on infants meeting the parameters that were set. The search result showed that Google Scholar reported only five items (see Table 1).

A switch to German was used for a similar search—Basal Stimulation AND infants. Using German keywords for a search in Google Scholar twenty-two sources were identified, out of which one was a qualification work and the rest were outside the thematic framework of the search. 

The search strategy was structured following the JBI Manual [16]. Furthermore, a search was conducted in selected available sources to find the existence of the individual components P, I, and O (O1 and O2). Subsequently, the Boolean operators used to combine the terms were used for the search.

**Table 1 children-10-00389-t001:** Results for P, I, O1, O2, and complex searches in the English, German, and Czech languages.

Database	P (n)	I (n)	O1 (n)	Complex O1 (n)	O2 (n)	Complex O2 (n)	Complex O1 German	Complex O2 German	Complex O1 Czech	Complex O2 Czech
PSYCINFO	7950	1	34 (C)	0	18,741	0	0	0	0	0
MEDLINE	1,523,406	33	18 (C)	0	1502	0	0	0	0	0
PsycArticles	6131	4	2 (C)	0	1009	0	0	0	0	0
ERIC	824	5	4 (C)	0	1327	0	0	0	0	0
Wiley Online Library	50,5083	7	8 (C)	0	14,462	0	0	0	0	0
ProQuest	878,955	5	53,086	0	27,279	0	0	0	0	0
Scopus	88,831	36	52,184	0	17,755	0	0	0	0	0
WOS	627,439	9	4711	0	18,549	0	0	0	0	0
JSTOR	703,776	14	41,065	0	949	0	0	0	0	0
Google Scholar	17,000	2340	352,000	1 Chakroun [17]	133,000	1 Vonikaki; Toumazani [18]	0	1 Sauseng-Prasser [19]	0	2 Baumruková [20] Ševčíková [21]
MedNar	635,656	68	146	0	90,310	0	0	0	0	0


***English*:**


P: preterms OR prematures OR multi-handicapped babies OR disabled OR premise OR premmies OR neonate OR newborn baby OR preemie OR newborn infant OR newborn OR preterm baby OR premie OR premature infant OR premature baby OR toddler OR infant

I: basale stimulation OR basic stimulation

O1: cognitive and behavioral functions

O2: temperament OR mettle


*The whole construction for target searching:*


O1 preterms OR prematures OR multi-handicapped babies OR disabled OR premise OR premmies OR neonate OR newborn baby OR preemie OR newborn infant OR newborn OR preterm baby OR premie OR premature infant OR premature baby OR toddler OR infant AND basale stimulation OR Basal Stimulation AND cognitive AND behavioral functions

O2 preterms OR prematures OR multihandicapped babies OR disabled OR premise OR premmies OR neonate OR newborn baby OR preemie OR newborn infant OR newborn OR preterm baby OR premie OR premature infant OR premature baby OR toddler OR infant AND basale stimulation OR Basal Stimulation AND temperament OR mettle


**
*Czech:*
**


O1 vícenásobně postižené dítě (multihandicapped babies) OR novorozeně (newborn infant) OR předčasně narozené dítě (premature infant) OR batole (toddler) OR kojenec (infant) AND bazální stimulace (Basal Stimulation) AND kognitivné a behaviorální funkce (cognitive and behavioral functions)

O2: vícenásobně postižené dítě (multihandicapped babies) OR novorozeně (newborn infant) OR předčasně narozené dítě (premature infant) OR batole (toddler) OR kojenec (infant) AND bazální stimulace (Basal Stimulation) AND temperament (temperament)


**
*German:*
**


O1: Mehrfach behindertes Kind (multihandicapped babies) OR Neugeborenes (newborn infant) OR Frühgeborenes (premature infant) OR Kleinkind (toddler) OR Säugling (infant) AND Basale Stimulation (Basal Stimulation) AND Kognitive und Verhaltensfunktion (cognitive and behavioral functions)

O2: Mehrfach behindertes Kind (multihandicapped babies) OR Neugeborenes (newborn infant) OR Frühgeborenes (premature infant) OR Kleinkind (toddler) OR Säugling (infant) AND Basale stimulation (Basal Stimulation) AND Temperament (temperament)

After the first experience of searching for relevant texts, when the results were very small in number, the time range of ten years from the publication of the text was extended to 15 years. The first step in searching was a localization study to select appropriate bibliographic sources. 

The first search in the databases below was targeted at a separate O1—the results were not thematically relevant to our study, for example, neurology, adults, neuropsychopharmacology, ADHD, central nervous system injuries, demented older adults, and chronic low back pain; the term that appeared most commonly was cognitive–behavioral therapy (CBT).

## 3. Results

Very few results were obtained when searching for sources using German variants of the keywords, as well as when using Czech. Because of the possible extension of the search results, an expert opinion was approached to search for texts. The use of a manual search on Google threw up seven sources and the website of the Institute of Basal Stimulation (Czech Republic) provided two sources in proceedings from conferences; the texts are in Czech and Slovak. Out of those sources, three expert communications were found. All of those were relevant to the topic that was the subject of the search. The articles that were found related only to the keywords Basal Stimulation AND newborn baby. The retrieved sources were subjected to a basic analysis (see Table 2) according to predetermined criteria: type of source, framework, research or sample, results, limitations of study, and implications for practice.

The above table shows that more than half of the sources found are qualification theses (bachelor’s, master’s, and doctoral). Of the nine qualifying theses, four are bachelor’s [22,27,30,31], four are master’s [19,20,25,29], and one is doctoral [17]. The remaining resources are one book [32], one technical article [23], and four conference papers [18,24,28,31]. 

For 12 of the studies, the target population was fully aligned with the research question. Thus, they were preterm infants. The three remaining studies [27,29,30] have a different target group—adults. After a consultation between all the authors (MP, PP, and JM), the studies remained under review. The reason for their inclusion is the careful description of basal stimuli—good practice that can be applied to preterm infants. A precisely described research sample can be found in only two studies [17,18]. In the other cases, it is only stated that the children were involved. However, the text then implies that these are infants or toddlers. Three studies [28,29,30] clearly declare a generally positive effect of Basal Stimulation on the development of preterm infants. Another four studies admit a positive effect of Basal Stimulation specifically on cognitive development or temperament [22,23,26,27]. In other cases, a description is given of Basal Stimulation in a given target group and how the individual developed as a result of Basal Stimulation. From this information, a positive effect can be inferred. Unfortunately, one cannot speak of relevant research based on these sources examined (see limits in Table 2). The last column in Table 2 shows that the sources examined can primarily be classified as examples of good practice. 

The results of the search indicate a space for research. For this reason, the studies found were then further evaluated in terms of potential further research. The following categories were created (see Table 3): BS benefits, pitfalls of the source, possibilities for deeper exploration, limits, potential for future research, and possible research questions for the future. All these categories indicate the current state of research on this problem and point to a possible way forward.

In the case of the pitfalls of the source, there are clear problems in terms of methodology. In six cases, the research methodology is not clearly described [19,22,23,24,25,26], in six cases, Basal Stimulation is described in very general terms or as part of a larger complex of therapeutic procedures [17,18,20,24,28,31], and in two cases, the case studies are from a single institution [27,29]. The last one [30] is a brief literary overview with a limited search strategy. In Table 3 (column 4) there is a description of the specific limitations. The last three columns then show the ways in which these specific studies could be expanded upon and the research mentioned deepened. 

## 4. Discussion

This literature review provides information on the analysis and summation of literary sources related to the research question mentioned above.

The search period was set at 15 years because of the lack of sources found. First, the key concepts P, I, C, O1, and O2 were searched for in the databases in several variants using a language thesaurus. For a separate P, a large number of resources were found in the databases—3,832,296. For a separate I, 2454 resources were sought, but it was not the concept of “Basal Stimulation” but different terms for basic stimulation. The individual key term O1 was found in 569,046 cases in databases. Finally, 1,272,934 occurrences were found for O2.

No relevant texts were found in the sources listed above. The results applicable to this review were found using Google Scholar. For a search phrase using O1, one dissertation was found with only an abstract in English and text in French, by Chakroun [16] m Qualité dyadique et régulation émotionnelle lors d’un court moment de séparation mère-bébé 12 mois après une naissance prématurée. Next, a phrase with O2 was used for the search. This phrase was matched by an article by Vonikaki and Toumazani [18]: A Greek Home Visiting Program of Early Child Intervention for Visually Impaired Infants and Preschoolers with and without Additional Disabilities: Present Reflections and Visions. 

A phrase search in German for O2 suited the text of the master’s thesis by Sauseng-Prasser [19], Die Bedeutung der taktilen Stimulation in der Kinderpflege am Beispiel der Babymassage, with a rating of B. Using a phrase search for O2, two qualifying works were found: Baumruková’s [20] use of the concept of “Basal Stimulation” in intensive care and Ševčíková’s [21] use of Basal Stimulation in health service providers. 

This was followed by another manual search using the Google platform, which used the entire range of expressions, in English, German, and Czech. A total of ten additional resources were found.

These 15 sources were then reviewed (see Table 2 and Table 3). The review concluded that there is no relevant research that addresses the effect of Basal Stimulation on temperament in preterm infants. All of the studies found contained various methodological flaws that make it impossible to view the results presented as conclusive. Basic methodological shortcomings include non-representative sampling [27,29] or an insufficient description of the methodology to determine the variability and reliability of the research [19,22,23,24,25,26]. In most cases, a theoretical elaboration of the concept of Basal Stimulation is lacking [17,18,20,24,28,31]. For this reason, the review focused on describing the shortcomings on the one hand and on the other hand, on the possibilities for developing research activities in this area. 

Basal Stimulation is a concept that is used as a full-fledged therapy for individuals with various types of disabilities [13,15]. Likewise, it appears to be an important support, specifically for premature babies [14,16]. The literature reviewed supports this on the basis of examples of good practice [19,21,23,24,27,29,30]. These studies [22,23,26,27] then admit a positive effect of Basal Stimulation, specifically on cognitive development or temperament.

Cognitive functions are among the basic functions of the brain that enable people to perceive things, people, and events around them, think, react to stimuli, or manage different tasks. Not only memory but also concentration, speech functions, speed of thinking, the ability to understand information, orientation, judgment, problem-solving, and planning—all these are cognitive functions [33]. Cognitive functions need to be developmentally trained through games, multitasking tasks, learning, activities that are presented, etc. The basic cognitive functions include memory—which is one of the most important functions of the brain. With memory, a person can work, learn, satisfy the basic needs of the body, and form social bonds. Executive functions are involved in planning, decision-making, and volitional processes. Another important function is speech, which enables communication, the ability to express oneself and to understand. The efficiency of work and learning is supported by the attention function. The function of spatial orientation is also important, enabling movement and the perception of dimensions. In premature and at-risk newborns, cognitive functions are already impaired from birth [34,35].

There are many different articles on the effect of temperament on the development of individuals [32,36,37,38]. One of them is an article by the present authors [38], which, although it deals with the relationship between temperament and school performance, also reviews definitions and characteristics of temperament. Temperament is a factor that can influence other phenomena from the point of view of psychology. The problem is with the definition of temperament. In definitions of temperament, three basic areas appear whereby the characteristics differ according to different authors (the use of a particular term, the scope of the definition of the term, and the content of the definition of the term) [38]. For this article, we pick up only the relevant definitions. For more on this problem, see [38]. Regarding the content of the term, Fung et al. [39] use the terms ‘regulation of emotions’ or ‘self-regulation’ [40]. Concerning the scope of its definition, the term ‘temperament’ refers only to the emotional characteristics of personality [41]. 

When it comes to the content of the definition of the term, in this case, it is a question of whether temperament can be considered as a genotypic basis of personality or if there is a phenomenological picture of it. Kagan et al. [42] state that this is constructed on the basis of biological factors that are congenital and may affect behavior. From this brief overview, it is clear that temperament can play a large role in an individual’s life. In premature babies, different areas are “suppressed” or delayed [43,44,45,46].

Basal Stimulation may thus be a way and an opportunity to develop temperament and cognitive function in premature babies. The studies reviewed here suggest this fact too [22,23,26,27]. The problem with these studies (as noted several times above) is that the research presented is not of sufficient scientific quality to answer the research questions in a generalizable way.

## 5. Conclusions

For this literature review the research question was “What effect does Basal Stimulation have on the cognitive and behavioral functions or temperament of a preterm or disabled infant?”. On the basis of a literature review of 15 sources, it is possible to answer that Basal Stimulation has a good effect on improving temperament in preterm or disabled infants. None of these studies were based on research carried out with evidence of exact results. Nevertheless, on the evidence of the literature sources that were studied, it can be concluded that at the level of professional communication, the importance of Basal Stimulation therapy in the field of supporting the development of cognitive functions and manifestations of temperament in premature newborns can be confirmed. 

The lack of relevant literature sources that deal with the research question highlights the need to carry out studies with a sufficiently robust research design. The results in the form of scientific evidence could support the widespread use of the Basal Stimulation concept in therapy aimed at groups of premature babies with weakened health conditions.

## 6. Recommendation for Future Research

Since no scientific studies have been found, our review shows the necessary direction for further research activities in this area. It will be necessary to examine the direct relationship between Basal Stimulation therapy, cognitive functions, and the temperament of immature newborns. The results could provide evidence for a methodology for evaluating the effectiveness of Basal Stimulation. Given the focus of other research projects, the following research topics may be proposed, regarding the information obtained in this study: 1. Basic research on the importance of Basal Stimulation. 2. Explore the areas in which Basal Stimulation is effective. The following research questions can then be formulated for these areas: 

1. Basic research on the importance of Basal Stimulation. What are the objective parameters for evaluating the effectiveness of BS therapy?

Which factors are independent variables when using the BS approach?Under what conditions can the BS concept be complementary to other rehabilitation techniques?Which conditions are important for increasing the erudition of healthcare professionals regarding the use of the BS concept?

2. Explore the areas in which Basal Stimulation is effective.

How can premature and multiply handicapped or disabled children benefit from the BS approach?How can the concept of BS affect the manifestations of temperament in children in their first year of life?Is there any limitation of the usage of the BS concept in the conditions of a clinic for newborns?

The proposed research lines have potential in both qualitative (grounded theory study) and quantitative designs (randomized controlled trials or experimental study).

## Figures and Tables

**Table 2 children-10-00389-t002:** Basic information about the sources found from Google search and Institute of Basal Stimulation.

Author/Title	Type of Source	Framework	Research/Sample	Results	Limitations	Implications for Practice, Research, and Theory
Neradová [22]	GS * bachelor’s thesis	Theory of use of Basal Stimulation in newborns and immature babies	No details about the sample	Influence of Basal Stimulation on temperament and cognitive behavioral functions	Only general, theoretical, no research report	Informative for workers in direct intervention
Bellusso, Desnos, and Segond [23]	GS article neuropsychiatrie de l’enfance et de l’adolescence, 62(2), 90–94.	Children with ASD features without mentioning age	Research—five children with ASD features; among other therapies Basal Stimulation was also used; the result is the overall development and improvement of cognitive functions	The effect of Basal Stimulation on temperament and cognitive functions	The age of the children is not indicated	Example of good practice
Kováčikova [24]	IBS * conference paper	Case study of Basal Stimulation in a child with FAS	Child with FAS	Overall development of a child with a specific syndrome with Basal Stimulation support	It does not have a scientific format; it is a communication of expert experience	Example of good practice
Baumruková [20]	GS master’s thesis	Knowledge of health professionals about Basal Stimulation	Research is directed to nurses and paramedics	General characteristics of neonatal development using Basal Stimulation	The topic is only marginally related to Basal Stimulation	Information for further education of health professionals
Vonikaki and Toumazani [18]	GS conference paper	Comprehensive stimulation of children with visual impairments	Children aged 0–6	Effect of comprehensive stimulation on the overall development of toddlers with visual impairments	1. Only children with visual impairments 2. The term Basal Stimulation is not used, but the situations and activities correspond to the concept	Informative for workers in direct intervention
Chakroun [17]	GS doctoral thesis	Premature birth stimulation, control of emotions, and cognitive and behavioral reactions	Comparative study of 17 premature babies with ten FT children, aged 25–33 weeks	Effect of comprehensive stimulation on the overall development of premature babies	The Basal Stimulation principle is not explicitly mentioned	The results of the study may form the basis for follow-up research; examples of good practice are presented for intervention
Sauseng-Prasser [19]	GS master’s thesis	Aimed at stimulating premature newborns	Research is not published, rather just a set of methodological recommendations leading to the development of the child; the case study is presented as an illustration	In general, about stimulating newborns	The Basal Stimulation principle is not explicitly stated; tactile and comprehensive stimulation with an effect on temperament is described	Example of good practice
Mayer [25]	GS master’s thesis	Premature babies	Research into the relationship between the temperament of immature newborns and maternal sensitivity. Describes the effects of care on temperament, as measured by IBQ and ICQ	Intervention in premature babies and their overall development; sensitivity of mothers	Basal Stimulation is not explicitly listed	Support for premature babies and maternal synergies
Behrman and Butler [26]	GS book, institute of medicine	Premature babies from a medical point of view	The authors present the conclusions of research on the topic of premature birth	Effect of comprehensive stimulation on temperament and cognitive functions	Comprehensive view of the issue of comprehensive stimulation in rehabilitation, without explicit introduction of the concept of Basal Stimulation	General information
Matysová [27]	GS bachelor’s thesis	Adults	Three case studies report on the development of cognitive functions and self-control using Basal Stimulation	The influence of Basal Stimulation on the cognitive functions and temperament of clients is confirmed	Case studies demonstrate the effective influence of Basal Stimulation	Example of good practice
Chovancová, Kaiserová, Šagátová, and Koňošová [28]	IBS conference paper	Information on Basal Stimulation and results in general	Premature newborns	Not specified; Basal Stimulation as an effective procedure to promote child development	It does not have a scientific format; it is a communication of expert experience	General information
Ševčíková [21]	GS master’s thesis	Basal Stimulation was used to improve cognitive functions and self-control	Key studies of three adult patients at an anesthesiology and resuscitation department	The concept of Basal Stimulation was recognized as a very effective approach	Target group only adults	Example of good practice; well-described theoretical background
Kováčiková [29]	IBS conference paper	Information on Basal Stimulation and results in general	Newborns with severe disabilities	Not specified; Basal Stimulation as an effective procedure to promote child development	It does not have a scientific format; it is a communication of expert experience	Example of good practice
Táborská [30]	GS bachelor’s thesis	Case studies of two adult patients with impaired consciousness; the investigation focused on the effectiveness of stimulation	Use of Basal Stimulation in occupational therapy	Research on development of perception using Basal Stimulation of two patients in poor health condition	The Basal Stimulation concept is used here to work with adult patients with severe health problems—intensive care clinic	Example of good practice
Borecká [31]	GS bachelor’s thesis	Comparison of usage of Basal Stimulation in seven neonatal departments, knowledge, and education of staff	Comparative study of hospital departments	Focus on the Basal Stimulation methodology	The text is intended more for medical management	Information for further education of health professionals

* GS (Google search), IBS (Institute of Basal Stimulation).

**Table 3 children-10-00389-t003:** Analysis of available literary sources.

Source	BS Benefits	Pitfalls of the Source	Limits	Possibilities for Deeper Exploration	Potential for Future Research	Possible Research Questions for the Future
Neradová, [22]	BS is a therapy that supports the development of cognitive functions and manifestations of the child’s temperament.	No research is described.	Low number of children from only one clinic.	Two independent randomized groups as a comparative study will improve the confirmation of the effectiveness of BS.	Prepare a thorough comparative study with a control sample from several different clinics.	Can the effectiveness of the impact of a BS approach on the development of cognitive functions and temperamental manifestations be confirmed?
Bellusso, Desnos, and Segond [23]	The study shows the effectiveness of BS therapy in terms of temperament and cognitive functions.	The description of the research is very superficial.	Target group—five children with ASD features. The age of the children is not indicated.	To complete the evaluation with the opinion of a physiotherapist and add evaluation of the control group.	A comparative study would show the effect of BS therapy on children with ASD, children with other problems, and intact children.	Which rehabilitation approach has the highest effectiveness in children diagnosed with ASD?
Kováčikova [24]	Confirmation of the effectiveness of BS therapy by a cardio specialist.	Monitoring and description of development are not mentioned; no research formats.	Very short communication; expert experience without detailed information.	Analysis of the effectiveness of the BS approach using case studies.	Randomized study from different clinics.	What are the objective parameters for evaluating the effectiveness of BS therapy in neonatal cardiology?
Baumruková [20]	Evaluating knowledge of health professionals about Basal Stimulation as a therapeutic concept.	The study is not focused on the effectiveness of the BS approach in newborn children. The effectiveness of BS is only mentioned marginally.	The target group and the basic goal of the research are limitations. It is not specifically related to the approach and effectiveness of the BS concept.	Extension of research to specialization training and experience with patients in different age groups.	Extension of the study to multiple clinics, considering experience with patients from different age groups and evaluation of BS courses in the light of these specifics.	Do BS specialization courses affect the effectiveness of this therapy in specific target groups of patients?
Vonikaki and Toumazani [18]	The relationship between BS and temperament and cognitive functions: the relationship is documented in a general view of comprehensive therapy.	The BS concept is not monitored as a separate rehabilitation technique.	Visually impaired infants and preschoolers with and without additional disabilities.	Remove BS elements from the stimulation rehabilitation complex and examine this effectiveness as a separate entity.	Prepare a research project aimed at using the BS approach with children with multiple handicaps.	How can multiply handicapped children with a primary visual impairment benefit from a BS approach?
Chakroun [17]	The text does not explicitly mention the concept of BS, but the authors describe the effectiveness of a comprehensive therapeutic approach focused on sensory and tactile stimulation.	Too broad a view of the stimulating therapeutic approach; the concept of BS must be found in the segments of therapies that are described.	The main focus of the research is on mother–baby separation.	A study comparing a broad all-in-one concept compared to the sole use of the BS concept.	Proposal for a research project aimed at using the approach of BS therapy in children and their response to the presence of the mother—separation syndrome.	Is there a possibility of using the BS concept in therapy in children of early ages with separation syndrome?
Sauseng-Prasser [19]	The effectiveness of the concept of comprehensive stimulation is documented, although the term BS is not used directly.	The research conditions are not described.	Attention is focused mainly on tactile stimulation and a broader framework of comprehensive stimulation, not especially on BS.	Remove separate BS elements and compare with the effectiveness of other techniques.	Research focused on a larger number of premature children and accurately described therapy compared to a control group without the application of the BS concept.	What is the level of effectiveness of tactile stimulation as part of the BS therapy as a comprehensive approach?
Mayer [25]	Case studies illustrate the effectiveness of stimulation based on the BS concept, but this is not explicitly mentioned here. Publications on BS are cited as basic literary sources.	Descriptions of the research are not provided.	A clear description of the research strategy and possible implementation.	Possibility of using standard research procedures and thus obtaining valid data.	Influence of the BS approach on the promotion of the development of temperament.	How can the concept of BS affect the manifestations of temperament in children in their first year of life?
Behrman and Butler [26]	The individual elements of the BS concept are described in the text. The concept of BS is not explicitly mentioned.	Research is described at different levels of quality.	More therapeutic segments are described in the research, but the concept of BS is not singled out.	Use one common research design to obtain comparable and eligible results.	The use of a uniform design will allow comparison of the results and the effectiveness of the BS concept approach in different clinics.	Which factors are independent variables when using the BS approach?
Matysová [27]	Case studies are used as examples of the possibility of supporting the development of cognitive functions and manifestations of temperament when the BS concept is used.	Low number of case studies from one special center.	The BS concept was applied in adult clients with combined disabilities.	Prepare multiple case studies with a view of different combinations of health handicaps and then describe the degree of effectiveness.	Prepare a research study that will work with patients’ personal and medical history to compare the effectiveness of the BS concept in adulthood.	What is the effectiveness of the BS concept in the different entry health conditions of adult patients in their childhood?
Chovancová, Kaiserová, Šagátová, and Koňošová [28]	Confirmation of the functionality of BS in the clinical environment in neonates.	General description of the use and effectiveness of the BS concept.	Formally, the text is processed as a rather general expert communication for a conference.	Describe more widely the entry conditions of the health condition of newborns and the process of therapy and its effectiveness.	The use of a uniform design will allow comparison of the results and effectiveness of the BS concept approach in different cases.	Are there any limitations on the usage of the BS concept in the conditions at the newborn clinic?
Ševčíková [21]	Confirmation of effectiveness in using the BS concept in intensive care in adult patients.	A small number of case studies coming from one specialized workplace.	Little coherence between the results of the survey among healthcare professionals and a small number of case studies.	If possible, carry out a deeper qualitative analysis of interviews with healthcare professionals.	Use this research design and apply to more clinical workplaces with more case studies and therapists.	Where is the limit of variability of external conditions and their impact on the effectiveness of the BS concept?
Kováčiková [29]	Confirmation of the effectiveness of the BS concept in neonatological cardiology.	General description of the use and effectiveness of the BS concept.	Limited number of cases.	To carry out a deeper qualitative analysis.	The use of a uniform design will allow comparison of the results and effectiveness of the BS concept approach in neonatal cardio patients.	Are there restrictive conditions for the use of the BS concept in newborns who are patients of a cardiology clinic?
Táborská [30]	General information on the effectiveness of BS as a part of a wider therapeutic approach.	A brief literary overview.	A not-so-perfect search strategy.	Use of some resources as a complement to an advanced literature review.	Research aimed at linking the BS concept with other therapies.	Under what conditions can the BS concept be complementary to other rehabilitation techniques?
Borecká [31]	An overview of the use of the BS concept in various workplaces.	The relationship between BS and temperament and cognitive functions is mentioned very generally.	Particular attention is paid to the description of management, knowledge of the concept of BS, and frequency of use in patients.	A bachelor’s thesis in this form does not allow for a deeper use of data.	Implementation of research focused on the level of competence of healthcare professionals for the use of the BS concept and possible specialization in terms of age categories and types of patients.	Which conditions are important for increasing the erudition of healthcare professionals concerning the use of the BS concept?

## Data Availability

No new data were created or analyzed in this study. Data sharing is not applicable to this article.

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
