# Peer review of "Basal Stimulation as Developmental Support in At-Risk Newborns: A Literature Review"

_children, 2023, doi:10.3390/children10020389_

Round 1

Reviewer 1 Report

The research addresses an important subject, with an appropriate methodology. It suggests some interesting future research lines.

The sources searched are the proper ones and the methodology of the search is correct.

On page 5 there are the terms of search in Czech and German. It would be advisable to add the translation to English in case they are not the exact terms as in English either indicate it is.

The evaluation ranking from A to C would need further explanation in the context of the article.

The analysis performed in Table 3 should be introduced by explaining its categories.

In the item 5. Recommendation for future research, it would be better to organize the questions posed regarding its subjects. I mean, they should not be presented one after the other without organizing them by topic.

Author Response

Thank you for your review. Your comments helped us improve the article.

Reviewer 2 Report

I consider that this literature review has several weaknesses that do not make me comfortable accepting it for publication.

From the outset, I consider that the title of this review could be more intuitive, letting us understand what  are the IV and DVs, and not referring to the variables as a sum of them (basal stimulation + cognitive-behavioural functions + temperament + premature and disabled babies), but reflecting here the objective/research question of the review study .

In addition, I believe that the introduction is not very clear about the concept of basal stimulation, which is a key concept in the article.

In topic 1.1 Premature and disabled babies, reference is made to the low current mortality in premature infants, but that in the case of premature infants below 28 weeks of age, the 50% who do not die, are impacted by many disabilities. However, the reference that follows this statement does not fully corroborate it, above all, they do not refer to the negative impact of prematurity on temperamento. Indeed, this study (8) states that premature infants do not have differences in terms of temperament in relation to those who are not premature, and that premature infants are more likely to have an easy temperament. Also, in one of the aforementioned studies, temperament is treated as an IV and not as an outcome variable (as in the present study), so that the reported result is not relevant to the research question posed in this review.

With regard to the impact of prematurity on cognitive functions (which is different from cognitive-behavioral functions), the paragraph dedicated to this topic is very brief and not very clear, focusing on the impact on health problems and less on cognitive functions – just one brief reference is made to the function of attention. It is also not very clear what kind of impact is this. There will certainly be much more literature on this topic in addition to the aforementioned study (8). Moreover, the concept of a disabled baby is not clearly defined.

In topic 1.2. the cognitive-behavioral model of human thinking is used as a theoretical background, assuming that cognition is a mental process for processing and interpreting information from the environment. It seems to me that there is some confusion here between cognitive functions and the cognitive behavioral model of human thought. However, assuming that it is the latter that you want to study, I wonder why you assume that this will have an impact on premature or disabled babies, who are the target population of your study? No rationale is presented for this hypothesis.

In this same topic 1.2. a review of the literature on these cognitive behavioral functions and temperament in preterm infants and not in the general population was expected. It is not clear what these cognitive-behavioural functions are for the authors. Temperament is defined but then it is not clear how premature and disabled children are different in terms of temperament, there is only a reference to a study that seems incipient to me and that refers only to premature and not to disabled children.

In the introduction nothing is said about the impact of basal stimulation on temperament and cognitive-behavioral functions in children born prematurely, which is the scope of this review.

With regard to the chapter of the results, I was waiting for a synthesis and integration of the results of the various articles/documents summarized in the table, which does not happen. It is just a presentation of the results of the search carried out.

With regard to the conclusions, I think it is abusive to consider that there is a positive effect of basal stimulation on cognitive-behavioural functions (which are not clearly defined in this review) and on the temperament of premature and disabled children, given that the documents that contribute to this conclusion are not empirical studies, they have weaknesses from the methodological point of view, they were not subject to peer-review, and the target population is not always the same as the one targeted on this review.

Author Response

Thank you for your review. The comments helped to improve the article.

Reviewer 3 Report

Congratulations on the current paper!

I am a neonatologist working in a tertiary center and I've seen many preterm babies.

The long-term complications are obvious and well-documented in many published papers. It is very important to enroll the preemies in rehabilitation programs and to prevent by any means any kind of complications.

At the National Conference Of Neonatology, I also presented two papers about neurological complications of prematurity and the role of rehabilitation for preemies. So I am familiar with the subject of your paper.

I consider it to be well-documented and should be accepted after minor writing corrections.

Author Response

Thank you for your review. Corrected by a native speaker and professional translator.